# Detection of a Reassortant H9N2 Avian Influenza Virus with Intercontinental Gene Segments in a Resident Australian Chestnut Teal

**DOI:** 10.3390/v12010088

**Published:** 2020-01-13

**Authors:** Tarka Raj Bhatta, Anthony Chamings, Jessy Vibin, Marcel Klaassen, Soren Alexandersen

**Affiliations:** 1Geelong Centre for Emerging Infectious Diseases, Geelong, Victoria 3220, Australia; 2School of Medicine, Deakin University, Geelong, Victoria 3220, Australia; 3Centre for Integrative Ecology, Deakin University, Victoria 3220, Australia; 4Barwon Health, University Hospital Geelong, Geelong, Victoria 3220, Australia

**Keywords:** avian influenza virus, low pathogenicity, Chestnut teal, Eurasian lineage, H9N2, phylogenetic analysis, reassortant

## Abstract

The present study reports the genetic characterization of a low-pathogenicity H9N2 avian influenza virus, initially from a pool and subsequently from individual faecal samples collected from Chestnut teals (*Anas castanea*) in southeastern Australia. Phylogenetic analyses of six full gene segments and two partial gene segments obtained from next-generation sequencing showed that this avian influenza virus, A/Chestnut teal/Australia/CT08.18/12952/2018 (H9N2), was a typical, low-pathogenicity, Eurasian aquatic bird lineage H9N2 virus, albeit containing the North American lineage nucleoprotein (NP) gene segment detected previously in Australian wild birds. This is the first report of a H9N2 avian influenza virus in resident wild birds in Australia, and although not in itself a cause of concern, is a clear indication of spillover and likely reassortment of influenza viruses between migratory and resident birds, and an indication that any lineage could potentially be introduced in this way.

## 1. Introduction

Avian influenza virus (AIV) has major direct and indirect impacts on public and veterinary health, as well as wildlife conservation. AIV is a single-stranded, eight-segmented, negative-sense RNA virus in the family *Orthomyxoviridae* [1]. Each of the gene segments codes for one or more proteins. The virus surface proteins, hemagglutinin (HA) and neuraminidase (NA), are encoded by two separate gene segments. The internal proteins, which include the polymerase basic proteins 1 and 2 (PB1 and PB2, respectively), polymerase acidic protein (PA), matrix proteins (M1 and M2), nucleoprotein (NP), and nuclear export protein and non-structural protein 1 (NEP-NS1), are encoded by the other six gene segments, respectively [2,3].

Wild ducks and other waterfowl, and to a lesser extent shorebirds and gulls, are considered the main natural reservoir of AIV [4]. AIVs are designated as a low-pathogenicity avian influenza (LPAI) virus or a highly pathogenic avian influenza (HPAI) virus, based on their ability to cause disease in young chickens. This is mediated via a motif of basic amino acids in the HA protein precursor, facilitating proteolytic cleavage by ubiquitous proteases and enabling more generalized tissue infection of HPAI [5,6,7]. To date, this change from LPAI to HPAI has occurred only in some H5 and H7 subtypes, although in principle it may be possible in other subtypes [8,9,10,11,12].

H9N2 AIV is prevalent in poultry and wild birds globally [13]. It has also been reported and isolated from mammalian species like swine [14,15] and humans [16,17,18], indicating an ability for interspecies transmission. Several studies have shown that some H9N2 viruses have acquired internal genes from highly pathogenic (HP) H5 [19,20] and H7 subtype viruses [21]. These H9N2 viruses may be at an increased risk for becoming a potential pandemic threat [22,23,24]. While Australia has no land borders, and intercontinental waterfowl migration is highly limited, millions of shorebirds migrate to and from the Australian continent, and may potentially carry with them a variety of AIV subtypes [25,26,27,28,29]. Resident dabbling wild duck species commonly found in wetlands and estuaries in coastal regions of south-eastern Australia, such as Chestnut teals (*Anas castanea*), share habitats with many migratory shorebirds [30,31,32], potentially resulting in an exchange of AIVs. [25]. The magnitude of this risk, however, is not known.

Phylogenetically, H9 HA genes are divided into four lineages: h9.1–h9.4. Lineages h9.1 and h9.2 include H9 AIVs isolated in North America. Lineage h9.3 (represented by A/duck/Hokkaido/26/99) contains avian H9 viruses mainly isolated from Eurasia, but includes some from North America. Lineage h9.4 (G1-like, Y280/G9-like, and Y439-like viruses) solely contains avian H9 strains from Eurasia, and includes viruses that have caused human infection [14,17,18,33]. Lineages h9.3 and h9.4 are further divided into sub-lineages [33].

The N2 NA genes have been classified into two lineages, n2.1 and n2.2, which include avian and mammalian influenza viruses, respectively, although viruses within avian sub-lineage n2.1.4 have caused human infections [16,17,33]. Lineages n2.1 and n2.2 are further divided into Eurasian and American sub-lineages [33]. The six internal gene segments have also been grouped as Eurasian and American, based on their lineages and sub-lineages [34]. Several studies in Australia have shown the presence of an H9 HA gene segment in combination with an NA gene segment other than N2 in wild birds [27,28,35]. Here we describe the first identification and characterization of an H9N2 AIV subtype in a resident wild duck species in southeastern Australia, and conclude that the virus is a typical aquatic bird LPAI virus, with seven gene segments from Eurasian lineage viruses and the NP gene segment from a North American lineage virus.

## 2. Materials and Methods

### 2.1. Sample Collection and Ethical Statement

Fresh faecal samples were collected from five wild-caught Chestnut teals older than one year in age on 13 August 2018 in Wallington, Victoria, Australia. These samples were frozen at −80 °C within 1–3 h of collection. The five faecal samples were initially processed for nucleic acid extraction as a pool, designated as pooled sample CT08.18. AIV was subsequently detected in two out of five cloacal and oropharyngeal swab samples collected from the same five individual birds by a matrix gene PCR performed at the World Health Organization (WHO) Collaborating Centre for Reference and Research on Influenza, Melbourne, Australia. The two corresponding individual faecal samples (samples 11356 and 12952) were later also processed individually, but after another 11 months of storage at −80 °C. The bird sample collection was approved under Deakin University’s Animal Ethics Committee project number B43–2016, as well as Department of Environment, Land, Water and Planning permit number 1008206.

### 2.2. Virus Enrichment, Nucleic Acid Extraction, cDNA Synthesis, and Non-Targeted Amplification of Pooled and Individual Samples

All samples, both pooled and individual, were processed, and virus particles were enriched using a protocol described previously [36,37]. Briefly, the samples were homogenised at 25 Hz for 2 min, followed by centrifugation at 17,000× *g* for 3 min, and the supernatant was filtered using a 0.8 µm polyethersulfone (PES) spin-filter. The samples were then divided into two, one aliquot ultracentrifuged at 178,000× *g* for 1 h and the other not ultracentrifuged. Both of the aliquots were then treated with benzonase and micrococcal nuclease for 2 h [36,37]. The ultracentrifuged samples were then treated with 200 µM PMAxx for 30 min in a PMA-Lit LED Photolysis Device, as indicated by the manufacturer (Biotium, Fremont, CA, USA). Finally, nucleic acids were extracted using the QIAamp Viral RNA Mini Kit (Qiagen, Hilden, Germany). The cDNA synthesis and DNA amplification were carried out using the SeqPlex RNA Amplification Kit (Sigma, St. Louis, MO, USA), as per the manufacturer’s protocol [36,37].

### 2.3. cDNA Synthesis and Targeted Avian Influenza Virus PCR Amplification from Individual Samples

cDNA was synthesized from the individual chestnut teal samples using 5 µL of nucleic acids (obtained after virus enrichment, as described above) as the starting material, using 10× PathAm FluA RT Enzyme Mix at 45 °C for 60 min, and then at 95 °C for 1 min, as described in the manufacturer’s protocol (Applied Biosystems, Foster City, CA, USA). Synthesized cDNA was then amplified using 25× PathAmp FluA PCR Primer Mix and 5 U/μL SuperTaq plus PCR Enzyme (Applied Biosystems) at 95 °C for 4 min, with 40 cycles of 95 °C for 15 s, 55 °C for 30 s, and 68 °C for 2 min, with a final 68 °C step for 7 min using an Applied Biosystems ProFlex PCR system (Thermofisher Scientific, Waltham, MA, USA). The amplified products were purified using 1.5× Agencourt AMPure XP kit (Beckman Coulter, Brea, CA, USA), and 5 µL of the purified products were then fragmented and amplified using the same SeqPlex Kit (Sigma), as described above, following the manufacturer’s protocol [36,37].

### 2.4. Library Preparation, Next Generation Sequencing (NGS), and Detection of Avian Influenza Virus Sequences

Library preparation was performed using the Ion Plus Fragment Library Kit and IonXpress Barcode Adapters 1–96 Kit (Thermo Fisher Scientific). The libraries were quantified using the Ion Library TaqMan Quantitation Kit (Thermo Fisher Scientific), and then pooled. Sequencing was performed using Ion 530 chips and an Ion Torrent S5XL System (Thermo Fisher Scientific). Generated sequence reads were compared to a local database of virus sequences downloaded from Genbank (December 2018) using BLASTN [38,39], with an e-value cut-off score of 1 × 10^−10^, and a BLASTX [38,39] query with an e-value cut-off of 1 × 10^−10^. After the initial detection of AIV in the pooled sample, the same libraries were run a second time, using an Ion 540 chip to increase the amount of sequence generated. Both the non-targeted and targeted amplified individual samples were sequenced using Ion 530 chips.

### 2.5. Alignment of Avian Influenza Virus Reads and Phylogenetic Analysis

The sequence data generated from pooled and individual samples were analysed for the presence of AIV sequences. The AIV sequence reads from non-targeted pooled and individual samples were aligned using MEGA 7 software [40], because of very few AIV reads present and partial gene sequences created for each of the eight different AIV genome segments. For the targeted AIV PCR-amplified individual samples, reads were mapped to the nearest reference genomes using the TMAP plugin [41] and a mapping quality score of 80 or higher. Consensus sequences were generated in Integrative Genomics Viewer software (IGV) (Broad Institute, Cambridge, MA, USA) [42]. The final consensus was run as a reference in the TMAP plugin [41] to determine an accurate read depth coverage.

Generated sequences were compared with the online NCBI Genbank database using BLASTN [38,39], and the closest, as well as additional related or representative sequences of significant clades from around the globe, were selected for further analyses. Selected sequences were aligned using Clustal-W [43] and phylogenetic trees, created using the Maximum Likelihood (ML) method with the best fitting model, as determined by MEGA 7 [40]. The robustness of different phylogenetic nodes was assessed using 1000 bootstrap replicates.

### 2.6. Attempt at Virus Isolation

Virus isolation was attempted on the two influenza PCR-positive individual samples in embryonated chicken eggs at the Australian Animal Health Laboratory (AAHL), Geelong.

## 3. Results

### 3.1. Detection of Avian Influenza Virus Sequences by Next-Generation Sequencing, Using the Non-Targeted Method

The pooled CT08.18 sample generated about 10 million reads in the first NGS run, of which 40 reads (corresponding to 0.0004% of total reads) mapped to AIV. Rerunning the same NGS libraries using a 540 chip generated another 31 million reads, of which 129 reads (0.0004% of total reads) mapped to AIV, resulting in a total of 169 reads and covering partial sequences of all eight gene segments of AIV from the pooled sample CT08.18 (Appendix A). These 169 reads came from the sample for which the enrichment included ultracentrifugation, consistent with the fresh, pooled sample containing complete AIV virions.

For the non-targeted individual samples, individual sample 11356 generated around 3.8 million reads, of which only four reads (0.0000014% of total reads) mapped to AIV. Individual sample 12952 generated around 13.7 million reads of which only 21 reads (0.0000015% of total reads) mapped to AIV. Interestingly, and in contrast to the pooled sample mentioned above, most of these AIV reads (19 of 25) came from the non-ultracentrifuged samples, indicating that AIV virions may have deteriorated during the prolonged storage. Only reads mapping to the NP gene segment were obtained from the individual sample 11356, whereas a very few reads (five for PB2, one for PB1, one for PA, three for HA, four for NP, five for NA, one for M1–M2, and one for NEP-NS1) were mapped to each of the eight gene segments in the individual sample 12952.

### 3.2. Detection of Avian Influenza Virus Sequences by Next-Generation Sequencing Using the Targeted Avian Influenza Virus PCR Amplification Method

In the two individual samples where the PathAmp Flu A influenza virus amplification was performed, individual sample 11356 generated around 73,000 reads, of which 1500 reads (2% of total reads) mapped to AIV; sample 12952 generated around 81,000 reads, of which more than 28,000 reads (~35% of total reads) mapped to AIV. Out of the eight AIV gene segments, we obtained partial sequences of six gene segments, except for PB1 and PA, in individual sample 11356, while for sample 12952 we obtained the full length sequences of these same six segments. Segments PB1 and PA were not found to be amplified in either of these samples. Similar to what we observed for the individual non-targeted samples, more AIV reads were obtained from the non-ultracentrifuged preparations, and only very few from the ultracentrifuged ones.

### 3.3. Phylogenetic Analysis and Comparison of Avian Influenza Virus Gene Segments

The six full consensus gene segment sequences from AIV targeted sequencing of individual sample 12952, together with concatenated PB1 and PA gene segment sequences from the pooled sample CT08.18, were selected for phylogenetic analysis. Detailed analysis of the obtained sequences and comparison with the reference sequences identified an H9N2 subtype AIV in our pooled and individual samples. The virus from the pooled sample was named as A/Chestnut teal/Australia/CT08.18/2018 (H9N2). Similarly, the virus from individual sample 12952 was named as A/Chestnut teal/Australia/CT08.18/12952/2018 (H9N2). Due to the few partial sequences obtained from sample 11356, results from this sample were not included in any phylogenetic trees. However, obtained partial AIV sequences from the amplified individual sample 11356 were found to be identical with the amplified individual sample 12952. Similarly, the limited partial sequences obtained from individual sample 12952 without influenza virus PCR amplification were found to be 100% identical to the 12952 influenza virus with PCR amplification. The respective gene segment sequences in the pooled sample were found to be almost identical (>99.5–99.9%) in five of the six segments, while the sequences for the matrix gene segment was found to be 100% identical.

### 3.4. Identification of Avian Influenza Virus Gene Segment Lineages

The highest nucleotide sequence identities for seven of the eight gene segments on Genbank were found to be Eurasian lineage AIVs from different species of wild birds. The NP gene segment showed 98.12% similarity to a North American lineage AIV, as shown in Table 1 and Table 2.

### 3.5. Phylogenetic Analysis of the H9 Gene Segment

A 1718 nt long, full consensus sequence of the H9 HA gene segment (Genbank accession: MN826601) was obtained from the individual sample 12952, with a coverage depth ranging from 3 to 1272 at a mapping quality threshold of 80 or higher. Phylogenetic analysis of this segment was performed together with a total of 44 representative H9 reference sequences selected from the NCBI GenBank database. A/Chestnut teal/Australia/CT08.18/12952/2018 (H9N2) was most similar in nucleotide sequence (approximately 98%) to A/Grey Teal/Victoria/GT001/2017(H9N1) from Australia [28] (Genbank accession: MK213322), and belongs to Eurasian sub-lineage h9.3.3 [33,44], as shown in Figure 1. Five other H9 gene segment sequences previously reported from Australian birds from 2007–2012 also belong to sub-lineage h9.3.3, although these were only ~91–93% identical to A/Chestnut teal/Australia/CT08.18/12952/2018 (H9N2). The HA cleavage site of A/Chestnut teal/Australia/CT08.18/12952/2018 (H9N2) was found to posess the amino acid sequence ASDR/GLF, confirming this as a low pathogenicity strain.

### 3.6. Phylogenetic Analysis of the N2 Gene

A 1466 nt long, full consensus sequence of the N2 NA gene segment (Genbank accession: MN826603) was obtained from the individual sample 12952, with a coverage depth ranging from 2 to 2585 and at a mapping quality threshold of 80 or higher. Phylogenetic analysis of this segment was performed, together with a total of 37 N2 representative reference sequences selected from the NCBI GenBank database. A/Chestnut teal/Australia/CT08.18/12952/2018 (H9N2) was most similar in nucleotide sequence (>98%) to A/duck/Mongolia/154/2015 (H1N2) (Genbank accession: LC121278), and belongs to the n2.1.6 Eurasian lineage [33,44], as shown in Figure 2. The N2 sequences from the American lineages formed separate clusters. To the best of our knowledge, this is the first report of an N2 gene sequence from an Australian bird. The other reference sequences included in Figure 2 were found to be ~85–98% identical to A/Chestnut teal/Australia/CT08.18/12952/2018 (H9N2).

### 3.7. Phylogenetic Analysis of the Nucleoprotein Gene

A 1550 nt long, full consensus sequence of NP gene segment (Genbank accession: MN826602) was obtained from the individual sample 12952, with a coverage depth ranging from 3 to 703, at a mapping quality threshold of 80 or higher. Phylogenetic analysis of this segment was performed, together with a total of 36 NP representative sequences selected from the NCBI GenBank database. A/Chestnut teal/Australia/CT08.18/12952/2018 (H9N2) was most similar in nucleotide sequence (>98%) to A/RuddyTurnstone/MW02/Tas/2014(H10N8) [28] from Australia (Genbank accession: MH453824). A/RuddyTurnstone/MW02/Tas/2014(H10N8) [28] belonged to sub-lineage S5.1, the avian NP lineage from North America from the 1960s–2000s [34,44,47] (Figure 3). The NP gene segment from other wild birds from Australia from 2012–2017 also fell in the same lineage S5.1, with around 97–98% nucleotide identity to A/Chestnut teal/Australia/CT08.18/12952/2018 (H9N2). In contrast, the NP gene from Australian wild birds from 1983 made a separate cluster as lineage S5.7, which is separate again from the Eurasian lineage S5.2.1, as shown in Figure 3.

### 3.8. Phylogenetic Analysis and Molecular Characterization of the Remaining 5 Internal Gene Segments of the H9N2 AIV

Phylogenetic analyses of the three internal gene segments (PB2, M1-M2 and NEP-NS1) were conducted using the obtained full gene segment sequences with a coverage depth ranging from 2 to 4634 at a mapping quality threshold of 80 and above from the individual sample 12952. For the gene segments PB1 and PA, where we did not obtain full length sequences, concatenated sequences of 1381nt and 1130nt long, respectively, were assembled from the pooled sample CT08.18 for the phylogenetic analysis. Comparison of the partial sequences of the PB1 and PA gene segments obtained from the individual sample 12952 without PCR amplification to the PB1 and PA from CT08.18, found that they were 100% identical where they overlapped. Phylogenetic analysis indicated that all 5 gene segments fell within the Eurasian lineage with the PB2 gene segment in sub-lineage S1.2.4, PB1 in S2.2.1, PA in S3.2.7, M1-M2 in S7.2.5 and NEP-NS1 in the S8.2.1 sub-lineages [34,44,47] (Table 1 and Table 2 and Appendix A).

### 3.9. Attempt at Virus Isolation

The attempt for AIV isolation on the two influenza PCR positive individual samples, in embryonated chicken eggs at the Australian Animal Health Laboratory (AAHL), Geelong, gave a negative result.

## 4. Discussion

In this study, we detected and sequenced an aquatic bird Eurasian sub-lineage h9.3.3–n2.1.6 H9N2 LPAI virus from faecal samples collected from two resident Chestnut teals in southeastern Australia. Based on the sequences generated, we concluded that A/Chestnut teal/Australia/CT08.18/12952/2018 (H9N2) was an H9N2 AIV, with seven out of eight gene segments of Eurasian lineage and an NP gene segment of North American lineage. This AIV is therefore most likely a reassortant, with gene segments from viruses from two different lineages, indicating the intercontinental exchange of viral RNA. Another study of Australian resident wild birds recently demonstrated a similar type of intercontinental reassortment in H9N1 and H3N1 AIV [28]. Based on the overall results from our phylogenetic analyses, it appears that the six gene segments other than PB1 and NA have been introduced into Australian resident wild birds sometime in the past, while the Eurasian N2 gene segment (Figure 2) and Eurasian PB1 gene segment (Appendix A) occurred more recently, as they have only been detected in recent wild bird AIV surveillance (after 2015). The viruses we found here are therefore the progeny from two or more AIV reassortment events.

The 100% identity of the partial sequences of PB1 and PA gene segments from individual sample 12952 and pooled sample CT08.18 on the one hand, and the absence of sequences from these segments in individual sample 11352 on the other, support that the obtained partial, concatenated sequences of PB1 and PA from CT08.18 likely came from bird 12952 only.

The negative results of the virus isolation in embryonated chicken eggs from individual samples may have been because of poor sample quality, as the individual samples had been stored for a year when virus isolation was attempted; alternatively, it may be due to a low level of infectious viral particles present in the samples [49]. A low level of AIV in these samples is consistent with the low number of reads obtained even after virus enrichment (only 0.0004% of AIV reads in the pool), and even fewer AIV reads in the individual samples not subjected to specific AIV PCR amplification. As most of the AIV reads in the individual samples came from the non-ultracentrifuged samples, we speculate that AIV virions may have deteriorated/dried out during the prolonged storage of the individual faecal samples, and that these reads most likely came from partially protected AIV RNA/nucleoprotein complexes. Nevertheless, it is interesting to note that despite a likely low initial virus content and individual samples having been stored for a year, the protocol, including pre-amplification for full length AIV gene segment sequences using a commercial AIV PCR kit, yielded full-length sequences of six out of eight gene segments. Thus, this method of combined virus enrichment (excluding ultracentrifugation), followed by specific AIV amplification and NGS, may be of use for other wild bird AIV investigations, where initial NGS or virus isolation attempts have failed or provided limited sequence information.

Although it is reassuring that the virus detected is a typical aquatic bird Eurasian sub-lineage h9.3.3–n2.1.6 H9N2 LPAI virus [50,51], and not of the sub-lineage h9.4–n2.1.4, that has caused infection in humans [16,17,18], the presence of a reassortant H9N2 AIV in a resident wild bird, with some segments similar to viruses from countries remote to Australia, clearly indicates that these resident species are not isolated in regards to the global circulation of AIV. Therefore, any AIV lineage, including highly pathogenic viruses, could one day arrive.

## 5. Data Availability

Sequences generated for six full gene segments from the individual 12952 samples have been deposited in NCBI Genbank with accession numbers (MN826600–MN826605). Similarly, accession numbers MN156428–MN156434 represent the two partial, concatenated gene segments from the pooled sample CT08.18. The remaining sequences from the pooled sample CT08.18 have the NCBI Genbank accession number MN156424–MN156427 and MN156435–MN156445. Additional datasets analyzed in the paper can be made available from the authors upon reasonable request.

## Figures and Tables

**Figure 1 viruses-12-00088-f001:**
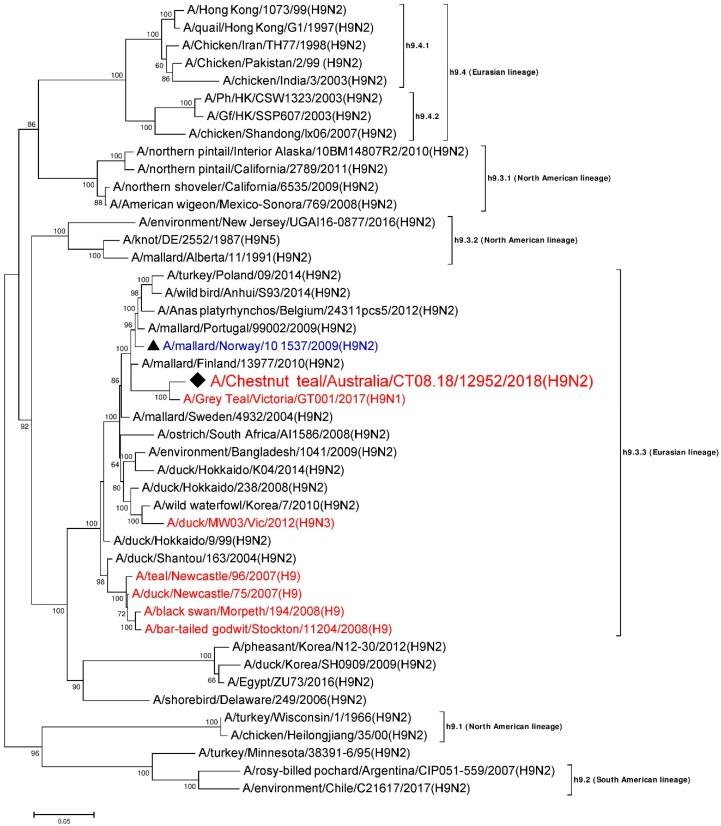
Phylogenetic analysis of the H9 gene segment. The nucleotide sequences were aligned and analyzed using the maximum likelihood method in MEGA 7.0 [40], using the Tamura-Nei (TN93+G) [45] model with a bootstrapping of 1000 replicates. The analysis involved 45 sequences of the H9 gene segment of influenza virus, including A/Chestnut teal/Australia/CT08.18/12952/2018 (H9N2) from a resident Chestnut teal. The numbers at the nodes represent bootstrap values, and only bootstrap values at or above 60% are shown. Branch lengths are scaled according to the number of nucleotide substitutions per site. The sequence from the current study has been labelled with a black rhombus (♦), while a representative reference sequence from Eurasian lineage has been labelled with a black triangle (▲) with text in blue color, and is shown on all figures for easy reference. Red colored sequences are from AIVs detected in Australia.

**Figure 2 viruses-12-00088-f002:**
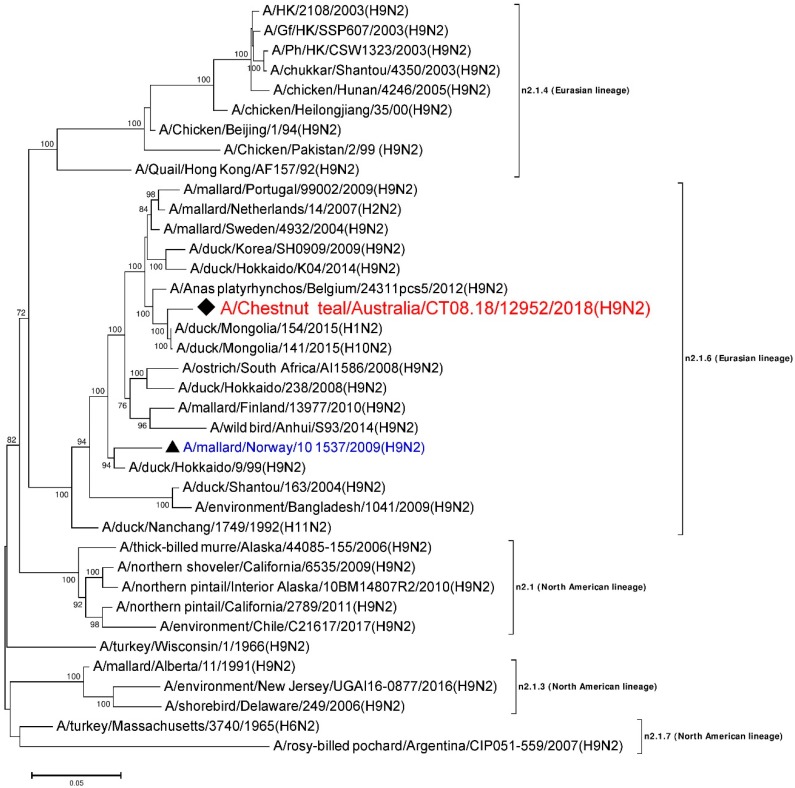
Phylogenetic analysis of the N2 gene segment. The nucleotide sequences were aligned and analyzed using the maximum likelihood method in MEGA 7.0 [40], using the Tamura three-parameter (T92+G) [46] model with a bootstrapping of 1000 replicates. The analysis involved 38 sequences of the N2 gene segment of influenza virus, including A/Chestnut teal/Australia/CT08.18/12952/2018 (H9N2), from a resident Chestnut teal. The numbers at the nodes represent bootstrap values, and only bootstrap values at or above 60% are shown. Branch lengths are scaled according to the number of nucleotide substitutions per site. The sequence from the current study has been labelled with a black rhombus (♦), while a representative reference sequence from Eurasian lineage has been labelled with a black triangle (▲), with text in blue color, and is shown on all figures for easy reference. Red colored sequences are from AIVs detected in Australia.

**Figure 3 viruses-12-00088-f003:**
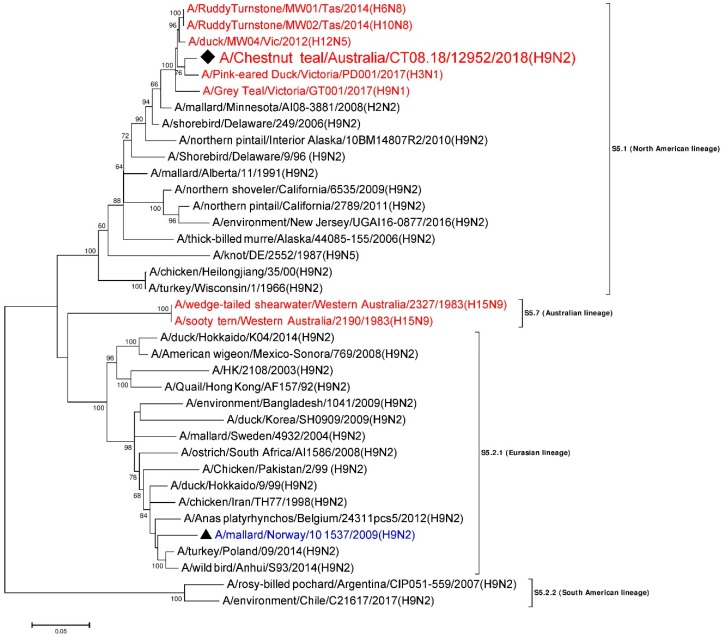
Phylogenetic analysis of the NP gene segment. The nucleotide sequences were aligned and analyzed using the maximum likelihood method in MEGA 7.0 [40], using the General Time Reversible (GTR+G+I) [48] model with a bootstrapping of 1000 replicates. The analysis involved 37 sequences of the NP gene segment of influenza virus, including A/Chestnut teal/Australia/CT08.18/12952/2018 (H9N2) from a resident Chestnut teal. The numbers at the nodes represent bootstrap values. Branch lengths are scaled according to the number of nucleotide substitutions per site, and only bootstrap values at or above 60% are shown. The sequence from the current study has been labelled with a black rhombus (♦), while a representative reference sequence from Eurasian lineage has been labelled with a black triangle (▲), with text in blue color, and is shown on all figures for easy reference. Red colored sequences are from AIVs detected in Australia.

**Table 1 viruses-12-00088-t001:** Avian influenza virus (AIV) isolates from GenBank with the highest nucleotide identities to complete gene segments of A/Chestnut teal/Australia/CT08.18/12952/2018 (H9N2).

Gene Segment	Genbank Virus Sequence with Highest Nucleotide Identity	Accession Number	Sub-Lineages *	Nucleotide Match	Identity %
PB2	A/duck/Bangladesh/26992/2015 (H7N9)	KY635525	S1.2.4 (Eurasian)	2280/2308	98.79
HA	A/Grey Teal/Victoria/GT001/2017 (H9N1)	MK213322	h9.3.3 (Eurasian)	1667/1703	97.89
NP	A/Ruddy Turnstone/MW02/Tas/2014 (H10N8)	MH453824	S5.1 (North American)	1517/1546	98.12
NA	A/duck/Mongolia/154/2015 (H1N2)	LC121278	n2.1.6 (Eurasian)	1441/1463	98.49
M1–M2	A/Grey Teal/Victoria/GT001/2017 (H9N1)	MK213323	S7.2.5 (Eurasian)	969/982	98.68
NEP-NS1	A/Ruddy Turnstone/MW02/Tas/2014 (H10N8)	MH453825	S8.2.1 (Eurasian)	872/882	98.87

*: “S” stands for sub-lineage; “h” and “n” stands for hemagglutinin and neuraminidase sub-lineages, respectively.

**Table 2 viruses-12-00088-t002:** AIV isolates from GenBank with the highest nucleotide identities to partial gene segments of A/Chestnut teal/Australia/CT08.18/2018 (H9N2).

Gene Segment	Genbank Virus Sequence with Highest Nucleotide Identity	Accession Number	Sub-Lineages *	Nucleotide Match	Identity %
PB1	A/duck/Mongolia/154/2015 (H1N2)	LC121274	S2.2.2 (Eurasian)	1338/1381	96.88
PA	A/Grey Teal/Victoria/GT001/2017 (H9N1)	MK213327	S3.2.7 (Eurasian)	1120/1130	99.12

*: “S” stands for sub-lineage.

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
