# Peer review of "Detection of a Reassortant H9N2 Avian Influenza Virus with Intercontinental Gene Segments in a Resident Australian Chestnut Teal"

_viruses, 2020, doi:10.3390/v12010088_

Round 1

Reviewer 1 Report

In this manuscript Bhatta and colleagues describe the first discovery of an H9N2 virus in birds in Australia. The virus appears to be an inter-continental reassortant comprising gene segments ofboth Eurasian and American origin.

The results of the paper comprise a single deep sequenced whole genome for which the authors do some fairly in depth phylogenetics, however no further virology is performed as infectious virus is unable to be isolated from the original samples.

The finding of an H9N2 in wild birds in Australia is new although, as the authors concede, H9 viruses (without N2) have been found in Australia previously. Additionally, as the authors acknowledge, the finding of inter-continental reassortants in Australia is not new (although these have been in different influenza subtypes).

The data in the paper is fairly limited – comprising of a single genome and some phylogenetics. The paper is concisely and well written including appropriate background, referencing and discussion.

We would recommend the following minor alteration:

Lines 56-60: could you please include the other common names for these H9 HA lineages (ie Y280/BJ94, G1 and Y439)

Author Response

Point 1: Lines 56-60: could you please include the other common names for these H9 HA lineages (ie Y280/BJ94, G1 and Y439)

Response 1: As requested by the reviewer we have updated the text of Lines 56-61 to now read as follows: Phylogenetically, the H9 HA gene segment sequences are divided into four lineages, h9.1- h9.4. Lineages h9.1 and h9.2 include H9 AIVs isolated in North America. Lineage h9.3 (represented by A/duck/Hokkaido/26/99) contains avian H9 viruses mainly isolated from Eurasia but includes some from North America. Lineage h9.4 (G1-like, Y280/G9-like and Y439-like viruses) solely contains avian H9 strains from Eurasia and includes viruses that have caused human infection [14, 17-18, 33]. Lineages h9.3 and h9.4 are further divided into sub-lineages [33].

In addition, as requested by this reviewer, we have made sure that proper spell checking has been done.

Reviewer 2 Report

The authors detect and describe of a reassortant H9N2 AIV in a pooled and individual fecal samples in a resident Austrian Chestnut Teal. The results showed that, one individual sample (12952) successfully generated six genome segments in NGS using both non-targeted and targeted AIV amplification method but failed to generate PB1 and PA segments. While, pooled sample rather managed to generate at least partial reads of all eight genome segments. Phylogenetic analysis revealed seven gene segments originate from Eurasian lineage whereas, NP gene related to North American lineage.

The studies are interesting, well designed and well written. Yet, there remain a number of minor issues that should be clarified:

Both individual fecal samples were stored at -80°C (Line 80-81) and were processed for virus particles enrichment (section 2.2). Further divided into two aliquot, one used for ultracentrifugation and other not. Finally, RNA was extracted from both aliquot and used for NGS. Further, the author stated in the section 3.1 line 155-157 (Interestingly, in contrast to the pooled………..deteriorated during the prolonged storage) provide confusion. It seems, failures to generate adequate reads form ultracentrifuged sample depends on the failure of ultracentrifugation method not for the sample deteriorated for prolonged storage at -80°c. This is the most recommended temperature for prolong virus storage. Line 158-159: ‘Partial sequences of all the 8 genome segments were found in the individual sample 12952’ is not it 6 genome segments from individual and 8 genome segments from pooled sample? Need clarification. Line 78 section 2.1: Same individual bird samples gives positive M gene PCR from swabs sample. Why then the NGS/ Sanger sequence not tried in swabs sample to compare the results of fecal samples? In the Table 1 and Table 2 at the lineages column S1, S2, S3…S8 was used but there was no footnote mentioning what stands for ‘S’. Although the author mentioned it on section 3.8 in line 280-281 but Table should be independent.

Author Response

Point 1: Both individual fecal samples were stored at -80°C (Line 80-81) and were processed for virus particles enrichment (section 2.2). Further divided into two aliquot, one used for ultracentrifugation and other not. Finally, RNA was extracted from both aliquot and used for NGS. Further, the author stated in the section 3.1 line 155-157 (Interestingly, in contrast to the pooled………..deteriorated during the prolonged storage) provide confusion. It seems, failures to generate adequate reads form ultracentrifuged sample depends on the failure of ultracentrifugation method not for the sample deteriorated for prolonged storage at -80°c. This is the most recommended temperature for prolong virus storage.

Response 1: We have addressed this point in the Discussion and updated the text of Lines 315-319 to now read: As most of the AIV reads in the individual samples came from the non-ultracentrifuged samples, we speculate that AIV virions may have deteriorated/dried out during the prolonged storage of the individual faecal samples and that these reads most likely came from partially protected AIV RNA/nucleoprotein complexes.

Our assumption is further supported by the finding of high number of reads of e.g. parvovirus, a highly stable virus in the same ultracentrifuged sample (not included in the paper but to be published separately) and from our otherwise highly successful protocol employing ultracentrifugation on other samples. Also, we would like to mention that the samples in question was very small aliquots of faecal samples, not swabs or suspensions in transport medium or buffer, and thus easily may have dried during storage. We have subsequently updated our protocols so that aliquots of faecal samples received in the lab are immediately prepared into 10% suspensions in transport medium before storage.

Point 2: Line 158-159: ‘Partial sequences of all the 8 genome segments were found in the individual sample 12952’ is not it 6 genome segments from individual and 8 genome segments from pooled sample? Need clarification.

Response 2: We have changed the text of Lines 159-162 to read: Only reads mapping to the NP gene segment was obtained from the individual sample 11356 whereas a very few reads (5 for PB2, 1 for PB1, 1 for PA, 3 for HA, 4 for NP, 5 for NA, 1 for M1-M2 and 1 for NEP-NS1) were mapped to each of the 8 gene segments in the individual sample 12952.

Point 3: Line 78 section 2.1: Same individual bird samples gives positive M gene PCR from swabs sample. Why then the NGS/ Sanger sequence not tried in swabs sample to compare the results of fecal samples?

Response 3: The swab samples were not accessible for processing at our laboratory as they were submitted, stored and tested in a separate laboratory as mentioned in the text. We only had the faecal samples to process, not the swabs from these birds. Hence, we used only faecal sample and were not able to compare NGS/Sanger Sequences between swab and faecal sample from the same individual.

Point 4: In the Table 1 and Table 2 at the lineages column S1, S2, S3…S8 was used but there was no footnote mentioning what stands for ‘S’. Although the author mentioned it on section 3.8 in line 280-281 but Table should be independent.

Response 4: In Table 1 after Line 196 and in Lines 197-198 and in Table 2 Lines 200 and 201we have made the following additions/changes: Changed ‘Lineages’ to ‘Sub-lineages* and added  *: ‘S’ stands for sub-lineage except ‘h’ and ‘n’ which stand for hemagglutinin and neuraminidase sub-lineages respectively.